# Gut site and sex-specific enrichment of bacterial taxa and predicted metabolic pathways in wild American black bear (*Ursus americanus*)

Erin A. McKenney[1]*, Erik De Jesus[2], Taylor Hatfield[3,¤a], Dorian Hayes[2,¤b], Kaleb Holder[4], Christian Ivarsson[2], Natalie Morais[2], Hunter Payne[2], Kalle Simpson[5], Adrianna M. Staal[6], Holly Thompson[2,¤c], Rachael Hildreth[7], Colleen Olfenbuttel[8], Diana J. R. Lafferty[7]

**1** Department of Applied Ecology, North Carolina State University, Raleigh, North Carolina, United States of America, **2** Department of Biological Sciences, North Carolina State University, Raleigh, North Carolina, United States of America, **3** Department of Horticulture, North Carolina State University, Raleigh, North Carolina, United States of America, **4** Department of Natural Resources, North Carolina State University, Raleigh, North Carolina, United States of America, **5** Department of Forestry and Environmental Resources, North Carolina State University, Raleigh, North Carolina, United States of America, **6** Department of Microbiology, North Carolina State University, Raleigh, North Carolina, United States of America, **7** Department of Biology, Northern Michigan University, Marquette, Michigan, United States of America, **8** North Carolina Wildlife Resources Commission, Wildlife Management Division, Raleigh, North Carolina, United States of America

☯ These authors contributed equally to this work.
¤a Current address: Tributary Land Design and Build, Durham, NC
¤b Current address: Director's Office, North Carolina Wildlife Resources Commission
¤c Current address: Henley on Thames, Oxfordshire, United Kingdom
* eamckenn@ncsu.edu

## Abstract

American black bears' (*Ursus americanus*) omnivorous feeding strategy, simple gut morphology, and rapid transit time prevent regulation of the gut microbiome (GMB). We analyzed stable isotopes and 16S rRNA sequences from 48 wild bears to assess the impacts of diet, age, gut site, and sex on GMB composition and PICRUSt2-predicted functional pathways. While alpha and beta diversity did not differ, we identified bacterial taxa and predicted pathways enriched based on gut site and sex. *Enterococcus, Incertae, Papillibacter*, and *Shuttleworthia* were enriched in jejunum samples (linear discriminant analysis effect size ≥ 3.5, p = 0.0374); and 6 genera drove colonic Bray-Curtis distances (SIMPER): *Weisella* (p = 0.0099), *Anaeroplasma* (p = 0.0495), *Megamonas* (p = 0.0099), *Cellulosilyticum* (p = 0.0495), *Escherichia-Shigella* (p = 0.0396) and *Ochrobactrum* (p = 0.0297). EdgeR identified isoflavonoid biosynthesis (p-adj = 0.001) and isoterpenoid biosynthesis (p-adj = 0.006) enriched in the colon, and SNARE interaction in vesicular transport (p-adj = 0.000) and secondary bile acid synthesis (p-adj = 0.005) enriched in females. Our findings provide nuanced insights to specific taxa and putative metabolic pathways that reflect sex and gut site differences in black bears, with important implications for understanding bear physiology and informing wildlife management.

**Data availability statement:** R code and data are available in the DRYAD repository (DOI: 10.5061/dryad.3tx95x6tb).

**Funding:** Funding was provided by the North Carolina Wildlife Resources Commission and grant funding (W57) through the Federal Aid in Wildlife Restoration Act, popularly known as the Pittman-Robertson Act. In-kind support was provided to DJRL and RH by the College of Arts and Science at Northern Michigan University. There was no additional external funding received for this study. The funders had no role in study design, data collection and analysis, decision to publish, or preparation of the manuscript.

**Competing interests:** The authors have declared that no competing interests exist.

## Introduction

Anthropogenic change is impacting diverse ecosystems throughout the world, and land use change in particular is profoundly affecting wildlife resource use [1]. Thanks to focused conservation efforts, the American black bear (*Ursus americanus*) population has experienced a dramatic recovery in North Carolina, from occupying an estimated 8% of the state in 1971 to over 60% in 2010 [2,3]. As black bears recover, their management becomes increasingly relevant. Increased agricultural land use in coastal regions like eastern North Carolina has led to more bears eating crops, and more black bears legally hunted by landowners and licensed hunters [2]. Understanding black bears' diets and associated gut microbiomes (GMB) can inform management both as a reference for health [4] and a reflection of anthropogenic impact [5,6].

Stable carbon ($\delta^{13}$C) and nitrogen ($\delta^{15}$N) isotope analyses derived from wildlife tissues, like hair, can be used to aid understanding of wildlife diets [7–9] including variation in trophic position among individuals within a population [10,11]. In fact, the ratio of stable carbon ($^{13}$C/$^{12}$C) and nitrogen ($^{15}$N/$^{14}$N) isotopes from ingested foods are incorporated into the consumer's tissues, creating an isotopic signature that can reveal information about what was eaten over the growth period of the tissues [12,13]. Because corn uses the $C_4$ photosynthetic pathway, and most bear forage plants use the $C_3$ pathway [14], the $\delta^{13}$C value for individual black bears provides insight into the variation in the proportional contribution of corn and other vegetation to the diet of black bears [15]. Similarly, the $\delta^{15}$N value derived from bear hair provides insight into trophic position, in which a higher $\delta^{15}$N value indicates more nitrogen in the diet, which may suggest the consumer is foraging at a higher trophic position (more animal matter in the diet) or consuming foods that have a high nitrogen content like peanuts [16].

Black bear diets in the eastern United States are dominated by berries, grasses, and hard mast [2,3,6]; but the availability of these crops has changed over time with increasing land conversion for agricultural use or development [17,18]. While the majority of black bears' generalist diets comprise plant material [19], black bears consume insects and other animals (e.g., domestic chickens, ungulates, fish) and retain the simple gut morphology of a true carnivore with a long small intestine, short, underdeveloped colon and no cecum [20]. Without a cecum, the black bear gut resembles a simple garden hose, lacking the morphological variation required to extend retention time sufficiently for GMB communities to differentiate between the small and large intestines [21]. The resulting rapid gut passage rates preclude regulation of the GMB compared to species with more complex guts, resulting in high inter-individual variation [21] that may further support black bears' generalist niche and adaptive flexibility [22], and reflect changes in landscape and foraging [23].

Published literature on carnivore gut microbiomes is limited (although see [15,21,23–26]) and we currently lack understanding of black bear GMBs and anthropogenic impacts on those GMBs in the southeastern United States. While recent studies have detected sexual dimorphism in farmed mink [24] and characterized functional differences in domestic versus feral cat GMBs [27], we do not fully understand the implications of sex-based differences in habitat use and foraging strategies

for black bear GMBs. To address these gaps, researchers at Northern Michigan University and the North Carolina Wildlife Resources Commission (NCWRC) partnered with students enrolled in an advanced quantitative course on Gut Microbial Ecology at NC State University. This cross-institutional collaboration not only addressed gaps in scientific understanding but provided a unique opportunity for students to apply foundational concepts to wildlife management through an authentic course-based research experience. In this study, we examined differences in stable isotope values, gut microbial community composition and predicted metabolic functions among 48 black bears legally harvested in eastern North Carolina (Table 1).

On average, male back bears travel further [28,29], which may expose them to more anthropogenic foods [30]. While one study documented males consuming more corn-based bait [31] than females, another study found no difference in male versus female δ13C stable isotopes values [32]. However, consumption of unprocessed and processed corn-based baits (i.e., corn, pastries, donuts, dog food) was previously shown to increase δ13C and decrease alpha diversity in black bears [15]. In the southeastern United States, 8 of 10 states with a regulated black bear hunting season (including North Carolina) allow licensed hunters to use baits to harvest a bear [33]. In North Carolina, hunters are restricted to using unprocessed foods (i.e., fruit, raw harvested agricultural crops) and prohibited from using bait that is comprised of any animal, animal part, or processed food products, which include any food substance or flavoring that has been modified or enhanced by sugar, honey, syrups, oils, salts, spices, peanut butter, grease, meat, bones, or blood; candies, pastries, gum, and sugar blocks; and extracts of such products [34]. The NCWRC estimates that 55% of bear hunters place bait for black bears and the most common bait type is shelled corn (65%) and raw peanuts (57%) [35]. Based on this information, we hypothesize that stable isotope values, alpha diversity, and predicted metabolic pathways will differ between male and female black bears. More specifically, we expect male black bears to exhibit enriched δ13C and harbor lower alpha diversity and distinct metabolic pathways as a result of their consumption of corn-based baits. Because males travel further for food than females, we also expect male black bears to host higher beta diversity associated with a more geographically varied diet.

By partnering with hunters, we have the unique opportunity to compare gut microbial communities across gut sites (i.e., small intestine, colon). Distinct differences in gut microbial diversity have been documented among gut sites across lemur species that have evolved specialized gut morphologies to support fermentation of fibrous diets [36]. By contrast, we hypothesize that, similar to a previous study of black bears in Michigan [21], we will detect no significant differences in either alpha or beta diversity across gut sites. However, we predict that the jejunum will host more differentially enriched taxa than the colon because GMBs located earlier in the gastrointestinal tract will have less time to undergo host or niche-specific selection.

## Results

Weight and age were significantly correlated in black bears (p < 0.00; S1 Fig). We therefore focused all subsequent analyses on age because it is tied to gut microbial succession [37].

**Table 1. Gut samples collected from black bears (*Ursus americanus*, n = 48) legally harvested in eastern North Carolina.**

|  | Number of samples |
| --- | --- |
| female | 16 |
| jejunum | 4 |
| colon | 12 |
| male | 32 |
| jejunum | 5 |
| colon | 27 |

## Stable isotope analysis

Overall, bears exhibited variable isotope values across intermediate trophic positions, typical for species with omnivorous diets [38,39], with no difference between males and females [40]. Several male and female black bears clustered between 6–8 δ15N and −15--10 δ13C, near the isotope values for corn [31] (Fig 1); however, no bears fell anywhere near the average isotope values for peanuts [41].

## Community composition

Firmicutes and Proteobacteria were the most dominant phyla across age classes, gut sites, and sexes, though male black bears hosted more *Proteobacteria (*17.2%) than females (13.6%). Similar phyla were present across samples, though *Fusobacteria* was unique to females, whereas *Acidobacteria* was only detected in males. Female black bears also hosted higher levels of *Bacteroidota*. At the family level, *Clostridiaceae, Enterobacteriaceae, Peptostreptococcaceae*, and *Streptococcaceae* dominated age classes (Fig 2A) and gut sites (Fig 2B). Of these, *Clostridiaceae* was most dominant (making up 39.8% of the bacteria in the colon and 32% in the jejunum; 46.8% in yearlings, 34.4% in subadults, and 37.5% in adults), driven by *Clostridium sensu stricto 1* (24.8% in the jejunum, and 25.1% in the colon; 32% in yearlings, 22.2% in subadults, and 24.3% in adults; Fig 2C-D). *Escherichia−Shigella* was also dominant, though more so in the jejunum

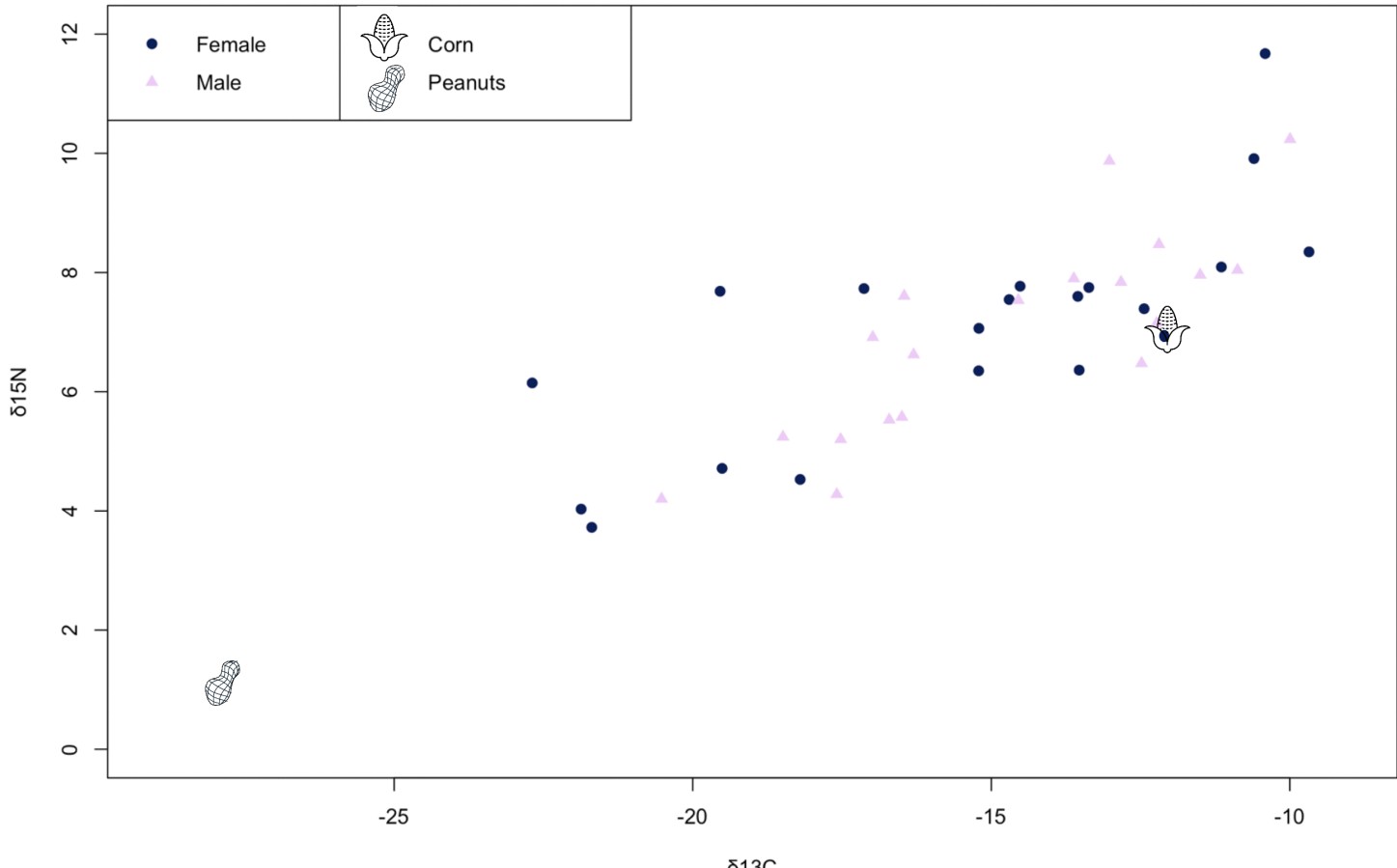

**Fig 1. Nitrogen and carbon stable isotope values for corn [42] and peanut [41] bait food reference values and male (n = 19) and female (n = 20) wild American black bears (*Ursus americanus*) harvested in eastern North Carolina.**

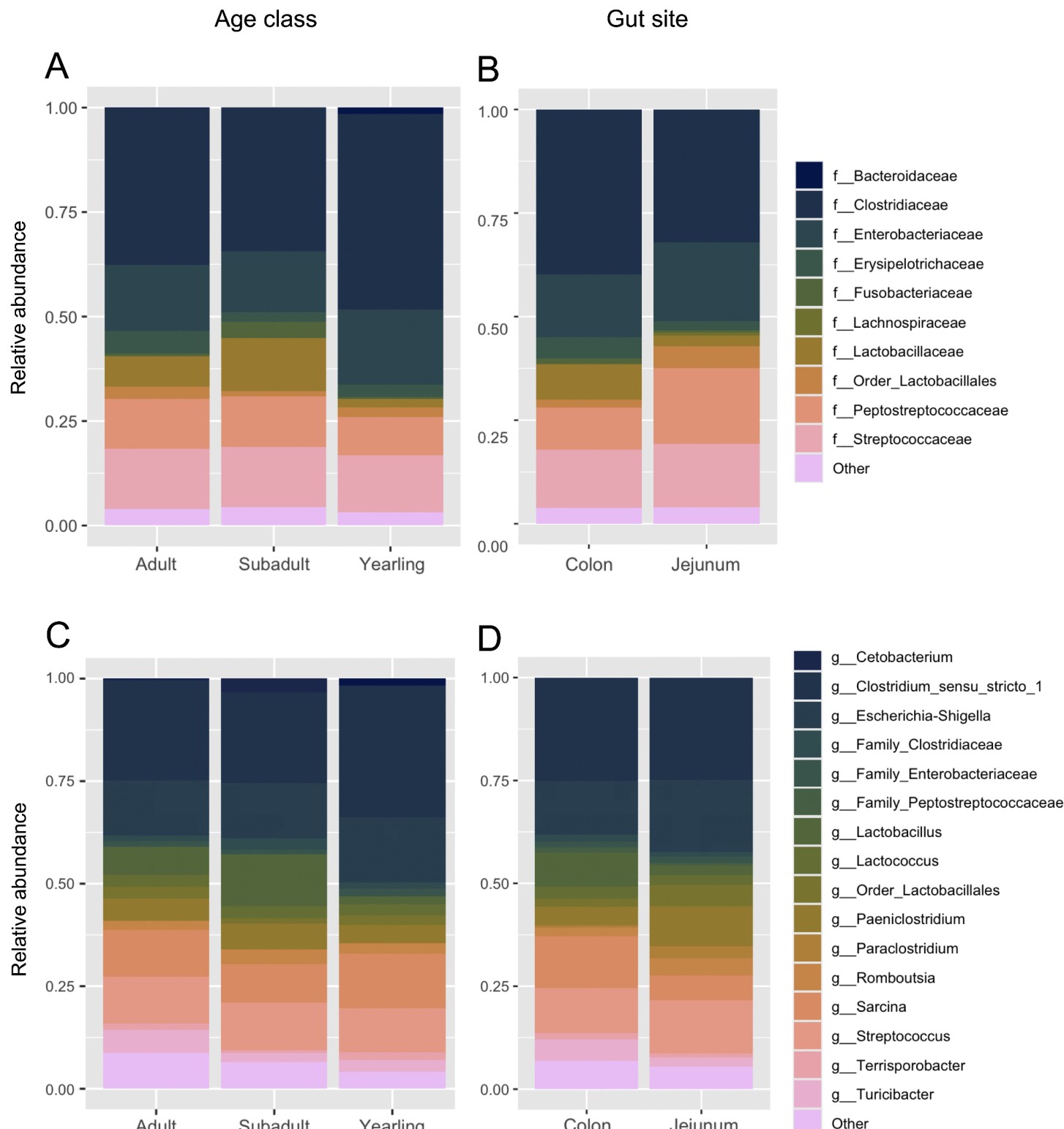

**Fig 2. Bar charts displaying the bacterial community composition across age classes (A,C) and gut sites (B, D) in male (n = 19) and female (n = 20) wild American black bears (*Ursus americanus*) harvested in eastern North Carolina.** Dominant taxa (present at >1% relative abundance) are classified at the family level (top 10, panels A-B)) and genus level (top 15, panels C-D).

(17.5%) than in the colon (13.0%). Lactobacillaceae: *Lactobacillus* was more abundant in subadults (12.7%, compared to 7% in adults and 2% in yearlings) and colon samples (8.3%, compared to 2.4% in jejunum).

In the jejunum, we observed the largest disparities between sexes for *Clostridium sensu stricto 1* (32.3% in females, 18.8% in males) and *Streptococcus* (18.9% in males, 5.5% in females) (Fig 3A). In the colon, females and males hosted similar levels of *Clostridium sensu stricto 1* but showed the largest disparity in *Turicibacter* (6.8% in males, 1.9% in females) (Fig 3B). Males and females hosted similar levels of *Escherichia−Shigella* across both gut sites.

By contrast, subadult bears hosted 50.2% *Eschericia−Shigella* in the jejunum (compared to 16.6% in adults and 3.7% in yearlings; Fig 4A), whereas yearlings hosted the most *Eschericia−Shigella* in the colon (20.7%, compared to 12.8% in adults and 8.1% in subadults; Fig 4B). *Clostridium sensu stricto 1* was most abundant in yearlings in both jejunum (30.4%, compared to 24.6% in adults and 15.1% in subadults) and colon samples (32.7%, compared to 24.2% in adults and 23.2% in subadults). Yearlings also hosted the most *Lactococcus* in the jejunum (8%, compared to 0.1% in subadults and 0.7% in adults), while subadults (3.3%) and adults (3.2%) hosted more in the colon (compared to only 0.6% in yearlings). Subadults and adults similarly hosted more *Lactobacillus* in both the jejunum (0.7% and 3.4%, respectively) and colon (14.4% and 7.7%, respectively) compared to yearlings (0.3% in jejunum and 2.7% in colon samples).

## Alpha diversity

Richness (but not Shannon diversity; S2 Fig) correlated significantly with age ($R^2 = 0.06142$, $p = 0.0494$; Fig 5), though Kruskal-Wallis tests detected no significant differences among age classes for either metric ($p = 0.3908$ and $p = 0.7433$, respectively; Fig 6A and 6B, Table 2). Male black bears hosted greater species richness at both gut sites (Fig 6C and 6E), while female black bears hosted greater Shannon diversity (Fig 6D and 6F). However, we detected no statistically significant differences in either richness ($p = 0.3347$) or Shannon diversity ($p = 0.6587$) between jejunum and colon, or between sexes within each gut site (Table 3).

## Beta diversity

ADONIS detected no significant differences in community composition based on age ($p = 0.923$) or between gut sites, regardless of whether sex was considered ($p = 0.56$ for full dataset and $p = 0.54$ for paired jejunum and colon samples from 9 bears) or not ($p = 0.508$). Bray-Curtis NMDS plots were characterized by high overlap, although inter-individual variation increased with age (Fig 7A and 7B) and paired gut samples exhibited less overall variation compared to the full dataset (Fig 7C and 7D).

SIMPER analysis identified 109 ASVs corresponding to 89 unique taxa that drove Bray-Curtis dissimilarities among age classes (S1 Table). Of these, 77 ASVs (57 genera) were enriched in yearlings compared to subadults; 18 ASVs (18 genera) were enriched in yearlings compared to adults; and 14 ASVs (14 genera) were enriched in subadults compared to adults. Notably, only two significant genera (*Romboutsia* and *Clostridiaceae*) qualified as major taxa (occurring at an average relative abundance of 1.37% and 1.06%, respectively); and both were enriched in subadults compared to adults.

We also detected 10 taxa that drove patterns in Bray Curtis distances for the colon, but none for the jejunum. We identified 8 significant taxa after combining redundant ASV classifications, 6 of which were identified to the genus level (Table 4). Specifically, *Weisella* ($p = 0.0099$), *Anaeroplasma* ($p = 0.0495$), *Megamonas* ($p = 0.0099$), and *Cellulosilyticum* ($p = 0.0495$) genera within phylum Firmicutes and *Escherichia-Shigella* ($p = 0.0396$) and *Ochrobactrum* ($p = 0.0297$) within phylum Proteobacteria were found to be significant.

LEfSe detected 71 distinct ASVs that were differentially enriched between gut sites with LDA effect size of ≥ 2.6. Only one ASV was differentially enriched in the colon: class *Vampirivibrionia* (LDA = 3.245036012, $p = 0.035$). In the jejunum, 10 ASVs had LDA effect size ≥ 3.5 and four ASVs could be classified to the genus level, all of which were within the jejunum: *Enterococcus, Incertae, Papillibacter*, and *Shuttleworthia* (Table 5). LEfSe also detected two unclassified genera in the family *Lactobacillaceae* that were enriched in females ($p = 0.0267$).

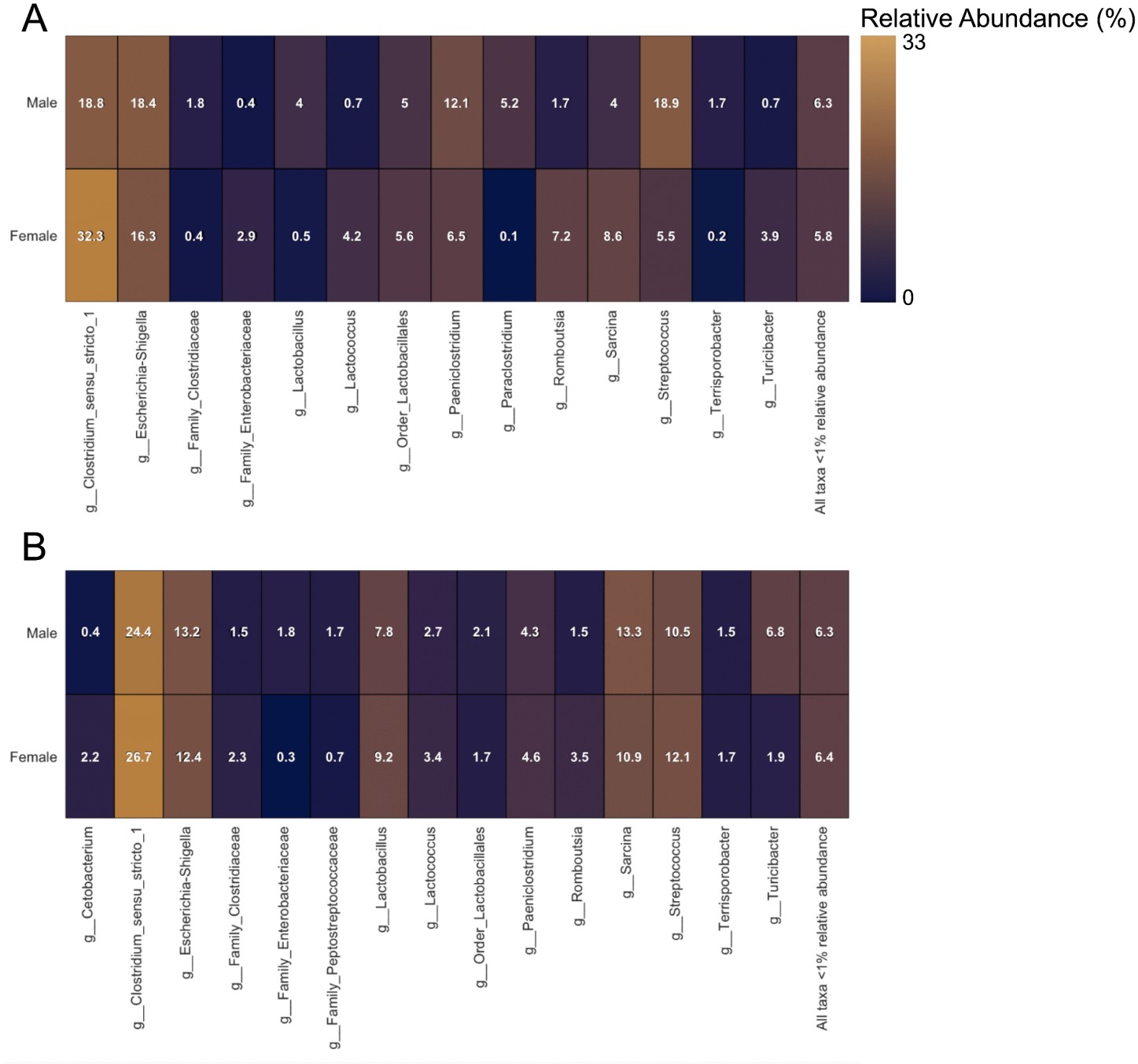

**Fig 3. Heatmaps of major taxa present at >1% relative abundance in in the jejunum (A; n = 9) and colon (B; n = 39), compared across sexes (n = 19 males and n = 20 females) in wild American black bears (*Ursus americanus*) harvested in eastern North Carolina.**

## Predicted metabolic pathways

We identified the top 15 predicted metabolic pathways based on their total abundance across all samples (Table 6). The top five were oxidative phosphorylation, purine metabolism, pyrimidine metabolism, neuroactive ligand-receptor interaction, and methane metabolism. Starch and sucrose metabolism was the only macronutrient pathway represented.

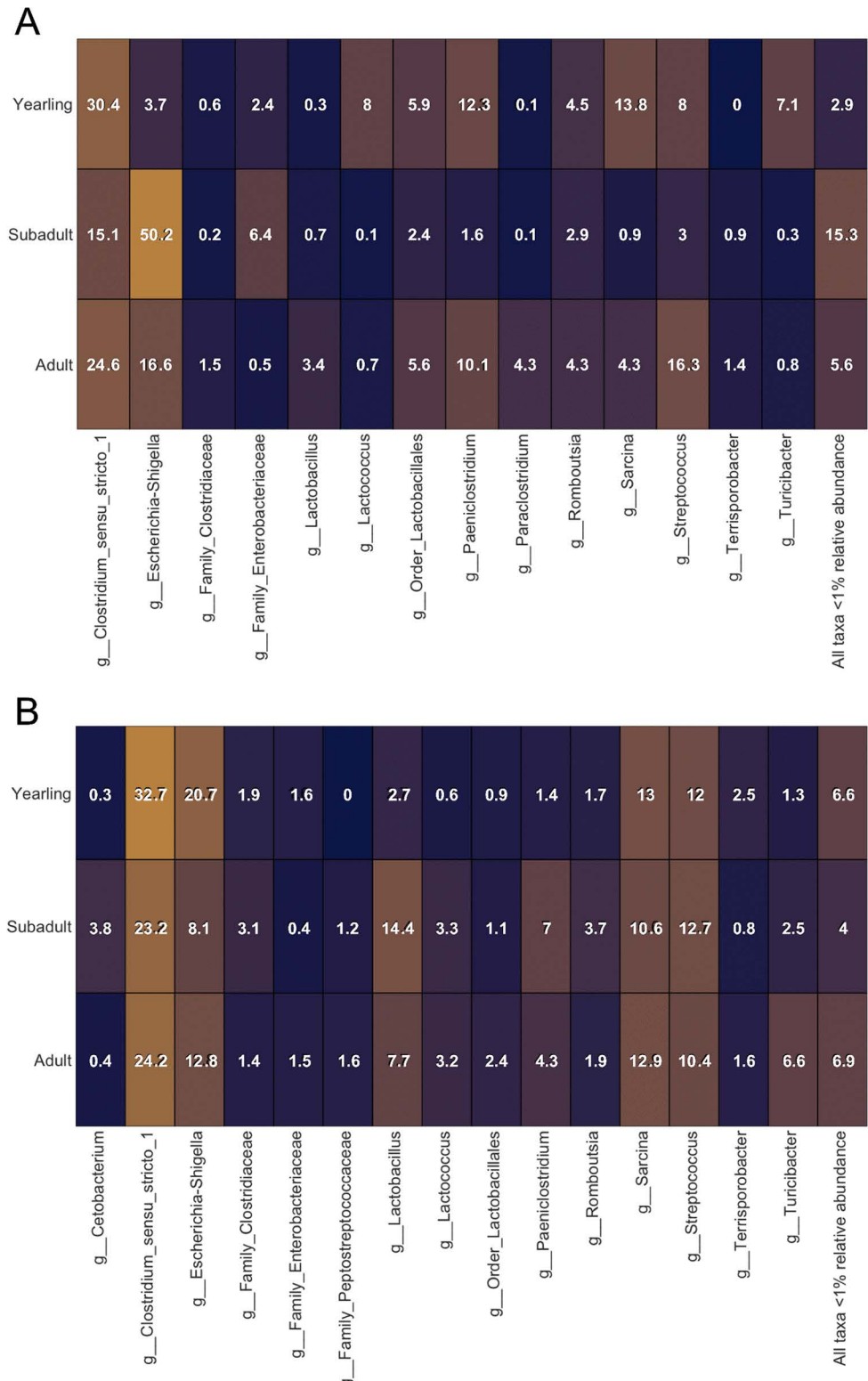

**Fig 4. Heatmaps of major taxa present at >1% relative abundance in in the jejunum (A; n = 9) and colon (B; n = 39), compared across age classes (C-D; n = 33 adults, n = 8 subadults; n = 7 yearlings), in wild American black bears (*Ursus americanus*) harvested in eastern North Carolina.**

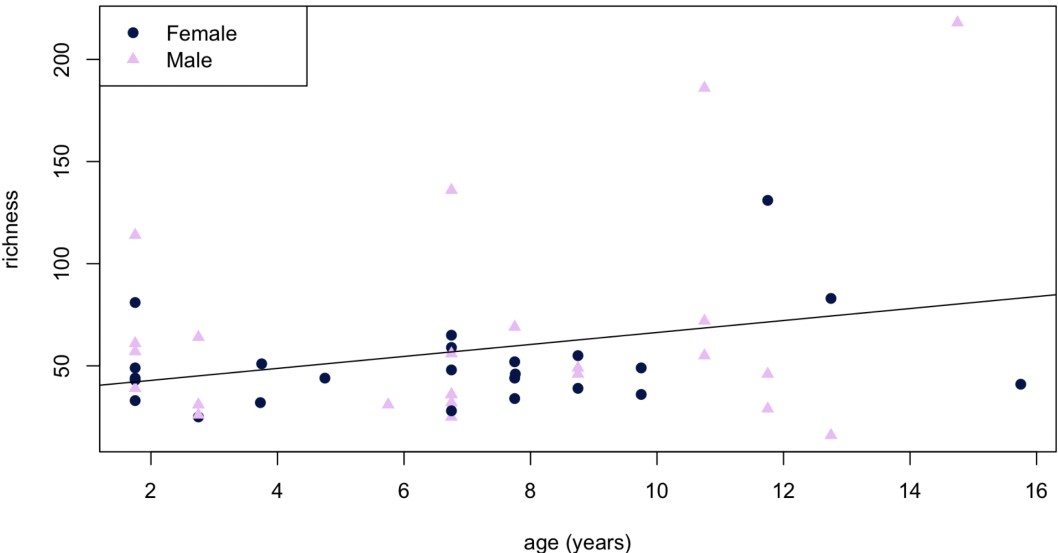

**Fig 5. Scatterplot of gut microbial richness versus age for 12 female and 27 male wild American black bears (*Ursus americanus*) harvested in eastern North Carolina.** The age of each individual was estimated by counting cementum layers from a canine tooth as described by Marks and Erickson [43].

**Table 2. Kruskal-Wallis test results comparing mean richness and Shannon diversity values across age classes (yearling, subadult, adult) in wild American black bears (*Ursus americanus*) harvested in eastern North Carolina.**

|  | richness | Shannon diversity |
|---|---|---|
| yearling | 59.4 | 2.20 |
| subadult | 41.8 | 2.06 |
| adult | 60.4 | 2.09 |
| chi-squared | 1.8792 | 0.59336 |
| p-value | 0.3908 | 0.7433 |

**Table 3. Mann Whitney U test results comparing mean richness and Shannon diversity values within gut sites (jejunum and colon) in male and female wild American black bears (*Ursus americanus*) harvested in eastern North Carolina.**

| Metric | Gut Site | Male | Female | W | p-value |
|---|---|---|---|---|---|
| Richness | Jejunum | 79.6 | 48 | 7.5 | 0.6228 |
| Shannon diversity | Jejunum | 2.17 | 2.23 | 12 | 0.7302 |
| Richness | Colon | 57.9 | 49 | 132.5 | 0.3773 |
| Shannon diversity | Colon | 2.07 | 2.11 | 7.5 | 0.6228 |

Principal component analysis (PCA) revealed distinct clustering patterns based on gut site: colon samples show greater functional diversity compared to jejunum samples (Fig 8A). PERMANOVA indicated no significant difference between colon and jejunum samples (p-value=0.394, $R^2$=0.0174). PCA revealed no clustering patterns for sex (Fig 8B), and PERMANOVA indicated no significant difference between male and female samples (p-value=0.897, $R^2$=0.0033). Variance analysis

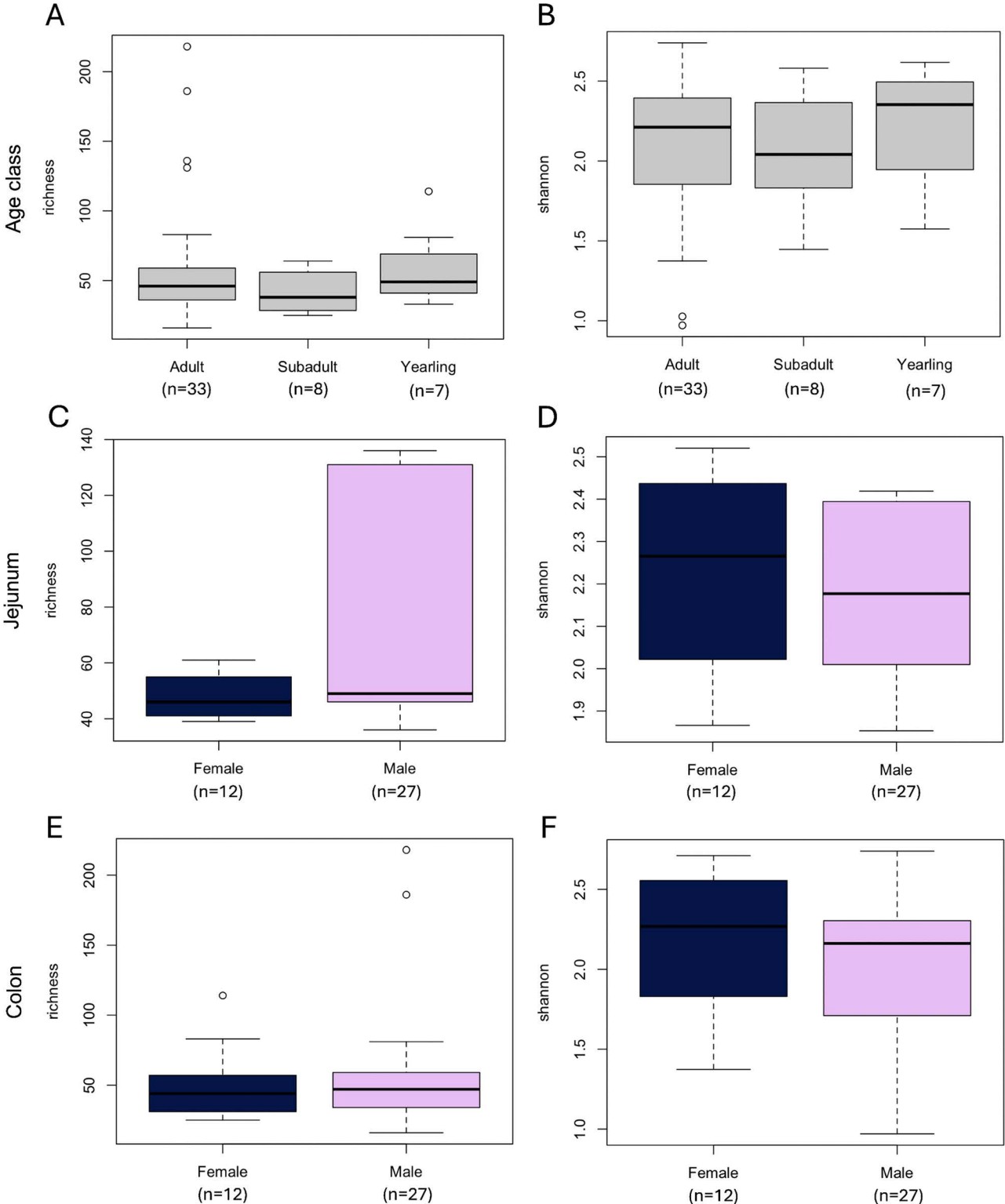

**Fig 6. Boxplots comparing alpha diversity across age classes (A-B) and gut sites (C-F) in 12 female and 27 male wild American black bears (*Ursus americanus*) harvested in eastern North Carolina.**

**Table 4. Colonic genera driving Bray-Curtis distances between gut sites, identified using SIMPER analysis (α = 0.05). Gut samples were collected from wild American black bears (*Ursus americanus*) harvested in eastern North Carolina.**

| Phylum | Class | Order | Family | Genus | Mean relative abundance | p-value |
|---|---|---|---|---|---|---|
| Firmicutes | Bacilli | Lactobacillales | Leuconostocaceae | *Weissella* | 1.65E-03 | 0.0099 |
| Firmicutes | Bacilli | Acholeplasmatales | Acholeplastmataceae | *Anaeroplasma* | 1.07E-06 | 0.0495 |
| Firmicutes | Negativicutes | Veillonellales-Selenomonadales | Selenomonadaceae | *Megamonas* | 6.33E-03 | 0.0099 |
| Firmicutes | Clostridia | Lachnospirales | Lachnospiraceae | *Cellulosilyticum* | 3.23E-04 | 0.0495 |
| Proteobacteria | Gammaproteobacteria | Enterobacterales | Enterobacteriaceae | *Escherichia-Shigella* | 1.30E-01 | 0.0396 |
| Proteobacteria | Alphaproteobacteria | Rhizobiales | Rhizoblaceae | *Ochrobactrum* | 3.20E-06 | 0.297 |

**Table 5. Mean relative abundance and linear discriminant analysis (LDA) effect size for significantly enriched genera in the jejunum, identified using LEfSe analysis (α = 0.05). Gut samples were collected from wild American black bears (*Ursus americanus*) harvested in eastern North Carolina.**

| Phylum | Class | Order | Family | Genus | Mean relative abundance | p-value | LDA effect size |
|---|---|---|---|---|---|---|---|
| Firmicutes | Bacilli | Lactobacillales | Enterococcaceae | *Enterococcus* | 5.01 | 0.0374 | 4.875 |
| Firmicutes | Clostridia | Oscillospirales | Ruminococcaceae | *Incertae* | 3.91 | 0.0374 | 3.724 |
| Firmicutes | Clostridia | Oscillospirales | Oscillospiraceae | *Papillibacter* | 3.78 | 0.0374 | 3.586 |
| Firmicutes | Clostridia | Lachnospirales | Lachnospiraceae | *Shuttleworthia* | 3.78 | 0.0374 | 3.569 |

showed that PC1 explained 74% of the total variance in predicted functional pathway abundances for both parameters, with PC2 explaining an additional 10%, and PCA3 and beyond explained less than 1% of the variance (S3 Fig).

Differential analysis with edgeR of KEGG relative abundances revealed two pathways that differed significantly between gut sites: isoflavonoid biosynthesis and isoterpenoid biosynthesis (Fig 9A, left). Both pathways remained significant upon Benjamini-Hochberg adjustment, with p-values of 0.001 and 0.006, respectively (Fig 9A, right). Between sexes, five pathways were identified as significantly different with edgeR analysis: SNARE interactions in vesicular transport, isoflavonoid biosynthesis, secondary bile acid biosynthesis, zeatin biosynthesis, and biosynthesis of type II polyketide products (Fig 9B, left). Of these, only two remained significant upon Benjamini-Hochberg adjustment: SNARE interactions in vesicular transport (p-value = 0.000) and secondary acid biosynthesis (p-value = 0.005) (Fig 9B, right).

The heatmaps in Fig 10 shows gut site specialization and sexual dimorphism in predicted microbial functions. The monoterpenoid biosynthesis pathway exhibited higher Z-scores across the jejunum samples compared to lower values in the colon samples (Fig 10A); the isoflavonoid biosynthesis pathway scored higher in the jejunum, on average, compared to the colon, which presented greater inter-individual variability among samples. In the sex-based heatmap (Fig 10B), female bears, on average, exhibited elevated Z-scores for SNARE interaction in vesicular transport and secondary bile acid biosynthesis compared to male bears. Males exhibited elevated values for type II polyketide products and zeatin biosynthesis, with zeatin biosynthesis especially pronounced. The isoflavonoid biosynthesis pathway exhibited high variability across samples.

## Discussion

The lack of clustering to any one bait food reference item (Fig 1) reflects that black bears are opportunistic feeders [19] and do not exhibit sex-specific foraging behaviors [40]. Similar to Gillman et al. [15,21], we found that *Firmicutes* and *Proteobacteria* dominate the black bear microbiome across gut sites and sexes. *Proteobacteria* has been previously identified as being involved in protein and carbohydrate digestion in dogs, and may be important for supporting black bears' omnivory. Similar to brown bears (*Ursos arctos*) [25], both male and female black bears hosted low relative abundances of

**Table 6. Top 15 predicted metabolic pathways across all samples collected from 48 wild American black bears (*Ursus americanus*) in eastern North Carolina. Metabolic pathways were predicted from 16S amplicon sequences using PICRUSt2.**

| Kegg ID | Pathway Name | Total Abundance |
| --- | --- | --- |
| ko00190 | Oxidative phosphorylation | 75306000.17 |
| ko00230 | Purine metabolism | 73471985.95 |
| ko00240 | Pyrimidine metabolism | 49960351.79 |
| ko04080 | Neuroactive ligand-receptor interaction | 42378600.14 |
| ko00680 | Methane metabolism | 40991842.13 |
| ko05010 | Alzheimer disease | 39493468.03 |
| ko05200 | Pathways in cancer | 38844446.97 |
| ko00910 | Nitrogen metabolism | 38423246.85 |
| ko04010 | MAPK signaling pathway | 37834076.3 |
| ko02020 | Two-component system | 36937803.85 |
| ko00620 | Pyruvate metabolism | 36654717.15 |
| ko05016 | Huntington disease | 36547375.91 |
| ko04610 | Complement and coagulation cascades | 34775358.88 |
| ko00500 | Starch and sucrose metabolism | 34195762.74 |
| ko05012 | Parkinson disease | 34125248.5 |

*Bacteroidetes.* While fiber-fermenting bacteria like *Bacteroidetes* are favored in omnivores and herbivores with a cecum, black bears and other taxonomic carnivores with simple gut morphologies are more likely to host facultative anaerobic bacteria adapted to metabolize protein and lipids and withstand rapid passage rates [21,23,26].

The lack of a significant difference in microbial diversity between the gut sites could be explained by bears' simple gut morphology (i.e., lack of cecum) [20], and the resulting rapid transit time and high digestive efficiency [44]. Many of the enriched and significant taxa that were identified in this study were found to be potential pathogens, some of which exhibit antibiotic resistance. For example, *Escherichia-Shigella*, *Clostridium sensu-stricto 1*, and *Weissella* were all found to be associated with infection in chickens [45]. Perhaps the short gut and rapid passage rates typical of carnivores act as a defense mechanism, passing potential pathogens before there is time for infection to occur. The presence (and potential dispersal) of antibiotic resistant microbes such as *Enterococcus* and *Ochrobactrum* could also have significant implications for humans and other wildlife [46].

*Escherichia-Shigella, Enterococcus, Cellulosilyticum,* and *Weissella* were previously identified as significant genera in Michigan black bears [21]. *Weissella* was also identified as a significant taxa in binturongs [47], another carnivore species with a simple gut morphology and omnivorous feeding strategy. It is notable that, despite the documented hypervariability across individuals within different carnivore species, we are still able to identify taxa consistently across different bear populations and even across different carnivore species.

We observed a greater average abundance of *Clostridium sensu stricto 1* in yearlings and female jejunum samples (Fig 3,4). *Clostridium sensu stricto 1* is a potential pathogen shown to contribute to disease development [45] and is considered an indicator of dysbiosis in human gut microbiomes [48]. *Clostridium sensu stricto 1* has also been identified as being positively and significantly associated with body weight and serum lipids in humans [49]. However, while obesity increases the risk of other chronic diseases in humans [50], microbially-facilitated weight gain may have more positive implications for key life stages in wildlife (e.g., growth in yearlings and reproduction in females). *Clostridium sensu stricto 1* is also a key taxon in dark fermentative production of hydrogen gas, a substrate used by *Firmicutes* to produce acetate, a highly

beneficial short chain fatty acid to peripheral host tissue [51]. This functional duality suggests that bacteria that cause disease in hosts with prolonged gut transit times may provide adaptive functions in species with simpler gut morphologies.

Like Gillman et al. [21], we detected no significant differences in alpha or beta diversity based on age. However, differences in microbial representation between age classes reflect classic succession dynamics predicted by Grimes' triangle [52], and likely result from changes in host regulation of the gut microbiome as the immune system develops. For example, opportunistic *Eschericia-Shigella* is more abundant in yearling colon samples, suggesting that it proliferates as it progresses through the gut. In subadults and adults, the taxon is less abundant in colon compared to the jejunum, suggesting that older bears' immune systems select against the potential pathogen. In addition to filtering, the immune system can promote more desirable gut taxa. For example, subadults and adults host more *Lactobacillus* in the colon, which are stress tolerant and contribute lactic acid toward host energy requirements [53].

While diet is also known to drive community composition in humans and other primate species [54,55], black bear gut microbiomes exhibit increasing richness (Fig 5) and inter-individual variation (Fig 7B) with increasing age, suggesting that hyper-omnivorous diets are not selective in species lacking a cecum. Furthermore, the statistical distinctions between age classes decrease as variation increases: SIMPER analysis detected 4 times more enriched taxa distinguishing yearling gut microbiomes from subadults (77 ASVs) than adults (18 ASVs) or between subadults and adults (14 ASVs; S1 Table). Together, these results emphasize the importance of gut microbial variation – including minor taxa – for imparting adaptive flexibility in omnivores [15].

PICRUSt2 analysis indicates that black bears host greater functional variation in the colon compared to the jejunum. However, only a fraction of the variation (1.75%) is attributable to gut site. This suggests that other factors may explain differences in functional profiles between gut sites, such as individual differences in diet, host physiology, or genetics. Together, the stable isotope scatterplot (Fig 1) and Fig 8 confirm that neither diet nor GMB function is predicted by sex.

Only 6 metabolic pathways were identified as significantly different across sex and gut site by edgeR analysis, a list that narrowed further (to 2 pathways per variable) upon Benjamini-Hochberg adjustment (Fig 9). These findings roughly aligned with previous findings [21], in that only two pathways were differentially abundant across both sex and gut site out of the hundreds identified. That said, these differences challenge our understanding of the carnivore "garden hose" gut, demonstrating that functional differences do exist across sex and gut sites despite the simplistic morphology of carnivore guts. Given that male black bears were previously found to produce higher and more variable levels of cortisol compared to females [40], it is possible that sex-specific differences in gut microbiome composition and functionality may be mediated by hormonal differences between males and females. For example, cortisol is associated with immune dysregulation, inflammation, and changes in gut physiology that disrupt gut microbiome composition [56].

Monoterpenoid showed the highest log2 fold change, whereas isoflavonoid log2 fold change was negligible (Fig 9, right). The relative abundance of monoterpenoid biosynthesis was greater in the jejunum than the colon. Monoterpenoids have industry applications ranging from cosmetic and medical. As such, research has largely focused on harnessing bacteria to produce these compounds [57]. Terpenes and terpenoids (the broader compound classes to which monoterpenoids belong) are considered to provide the anticancer, antioxidant, and anti-inflammatory benefits in essential oils [58]. Monoterpenoids have also been demonstrated to have antimicrobial properties against *Staphylococcus aureus* [59]. We suspect that these compounds might promote gut health in bears, but future research is needed to investigate the impacts of monoterpenoids on the GMB.

Secondary bile acid biosynthesis and SNARE interactions remained significantly enriched in females post-Benjamini-Hochberg adjustment (Fig 9, left) and have similar log2 fold changes (Fig 9, right). Secondary bile acids produced by the microbiome perform a variety of functions in the host. For example, urosdeoxycholic acid reduced weight in mice by altering both the microbiota and total bile acid composition [60]. Sex differences in secondary bile acid biosynthesis have not been characterized beyond mouse models [61] but we think they should be explored further, particularly given the importance of fat stores for supporting reproduction in ursids [62–64].

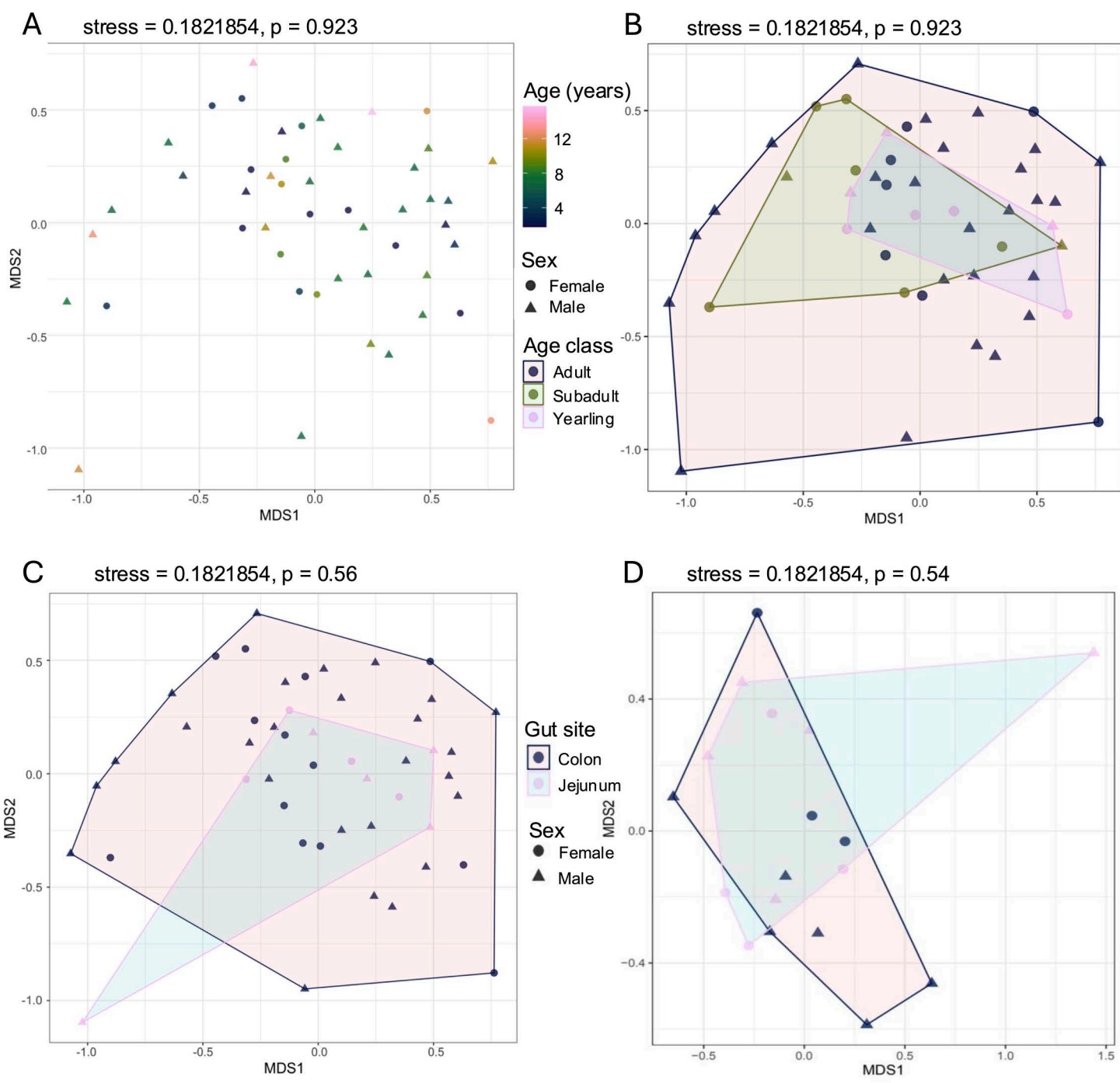

**Fig 7. NMDS plots visualizing pairwise Bray-Curtis distances among wild American black bears *(Ursus americanus)*, based on age (A-B) and gut site (C-D).** (A) The age of each individual was estimated by counting cementum layers from a canine tooth as described by Marks and Erickson [43]. (B) Using these data, we categorized bears into age-classes (yearling = 1, subadult = 2–3, adult ≥ 4) as previously described by Gillman et al. [21]. We compared gut sites for (C) all colon (n = 39) and jejunum (n = 9) samples and (D) paired samples from colon (n = 9) and jejunum (n = 9). All samples were collected from black bears harvested in eastern North Carolina.

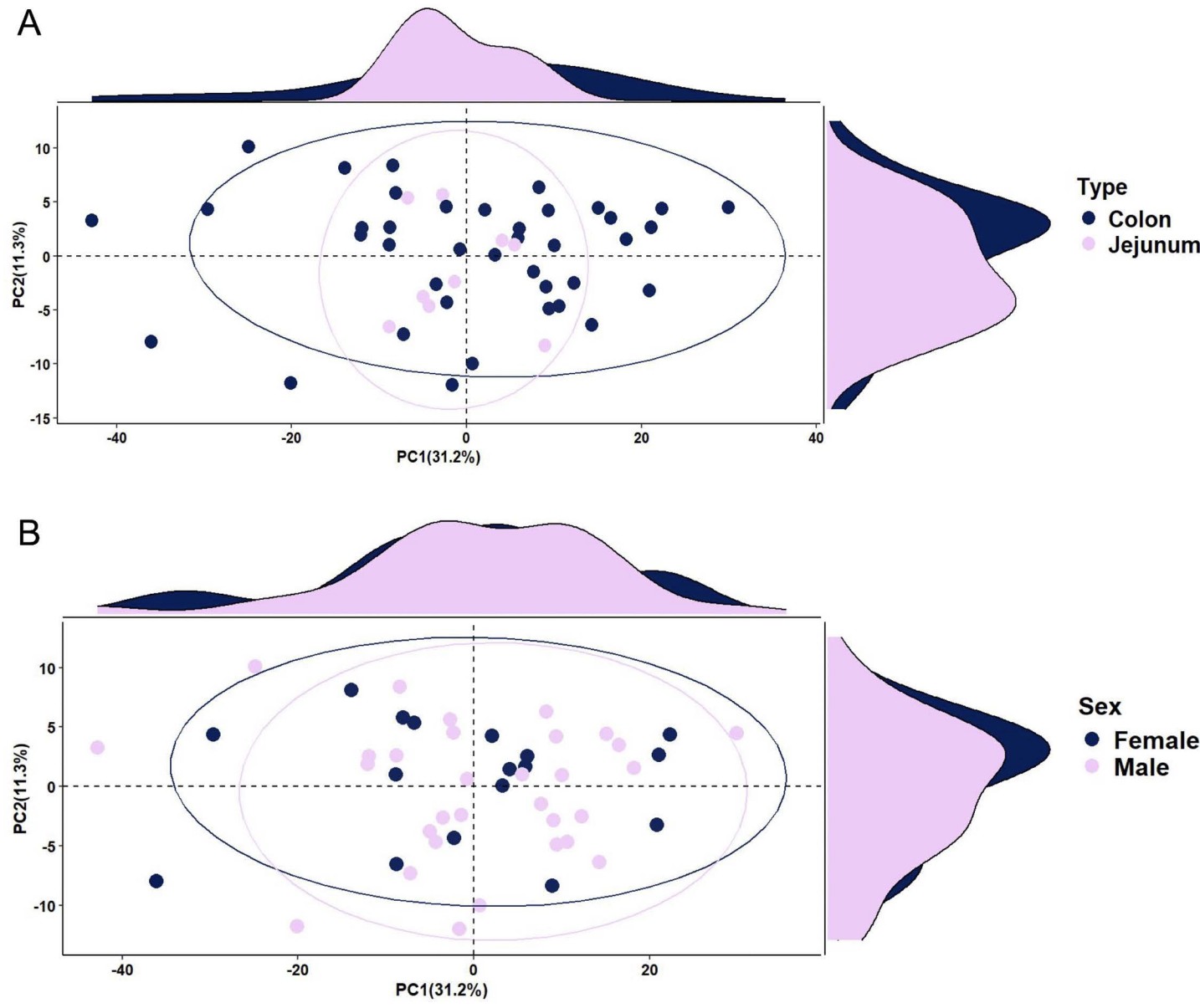

**Fig 8. Principal component analysis (PCA) plots visualizing differences in the abundance of predicted functional pathways between (A) gut sites (n = 39 colon and n = 9 jejunum samples) and (B) sexes (n = 20 females and n = 19 males).** Metabolic pathways were predicted from 16S ampli-con sequences for 48 wild American black bears (*Ursus americanus*) harvested in eastern North Carolina, using PICRUSt2.

A potential limitation to this study is the bias toward colon (39 versus 9 jejunum) and male (32 versus 16 female) samples, which may have prevented the detection of more significant differences associated with gut site and sex. A larger sample size might also decrease the masking effects of inter-individual variation. While PICRUSt2 is more accurate than its predecessor, it is not a replacement for standard shotgun metagenomics methods. Its use in the current study and conclusions drawn from its output should be interpreted as an estimation [65]. Further research is needed to determine how GMBs contribute to adaptive flexibility across a changing nutritional landscape. Specifically, a combination of metabarcoding and shotgun metagenomics could be employed to investigate how individual bears' diets may select for specific

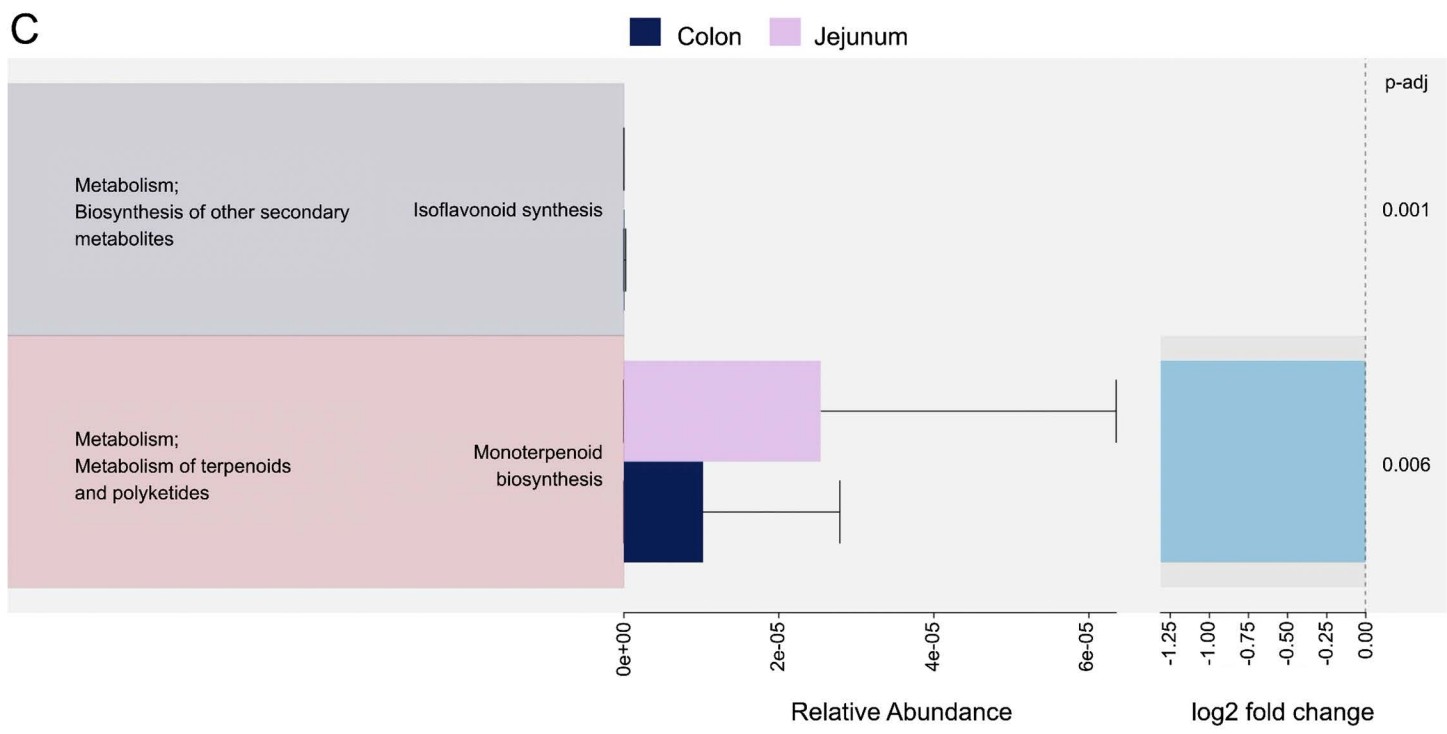

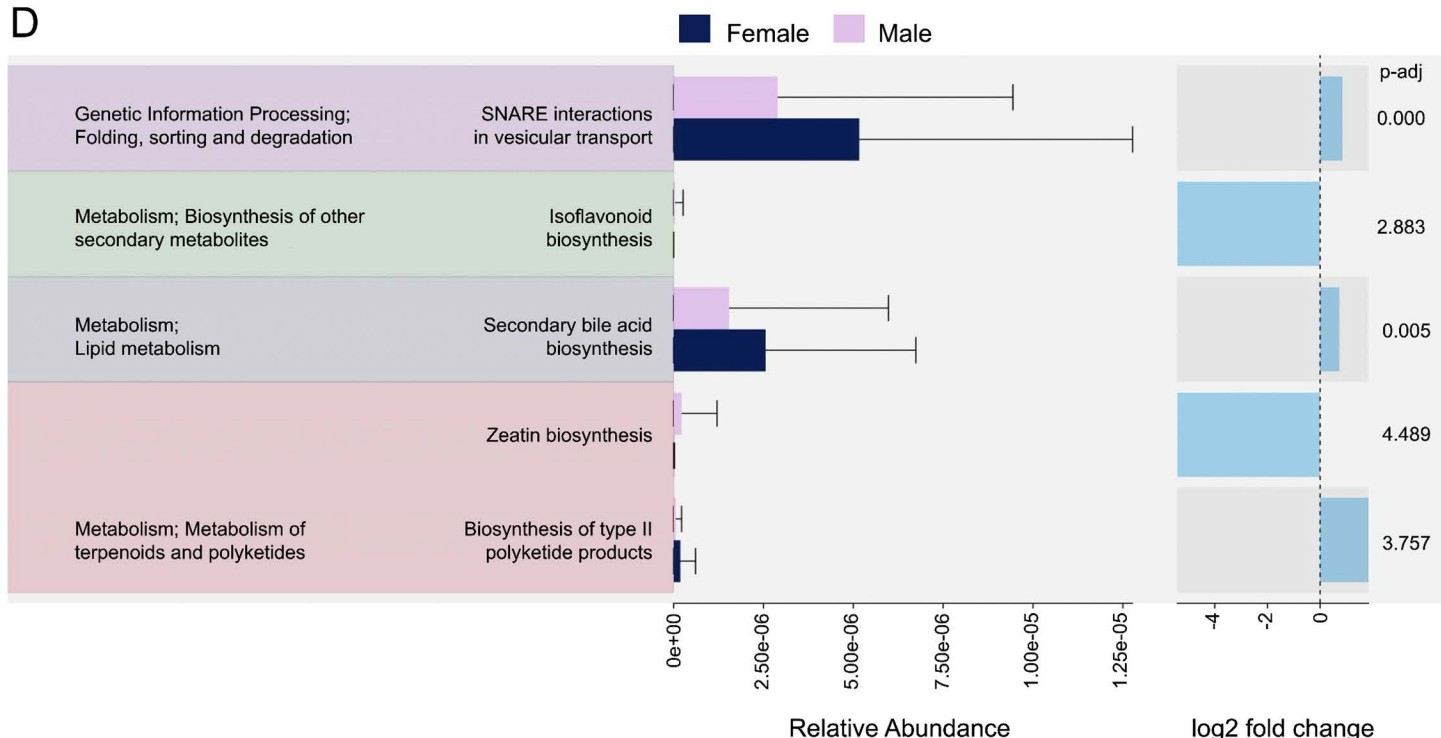

**Fig 9. Horizontal bar charts with error bars quantifying the relative abundance of the significantly enriched predicted metabolic pathways identified using edgeR differential analysis, with associated log2 fold change of pathway abundance with Benjamini-Hochberg adjusted p-values on the right, based on gut site (A) and sex (B).** Metabolic pathways were predicted from 16S amplicon sequences for 48 wild American black bears (*Ursus americanus*) harvested in eastern North Carolina, using PICRUSt2.

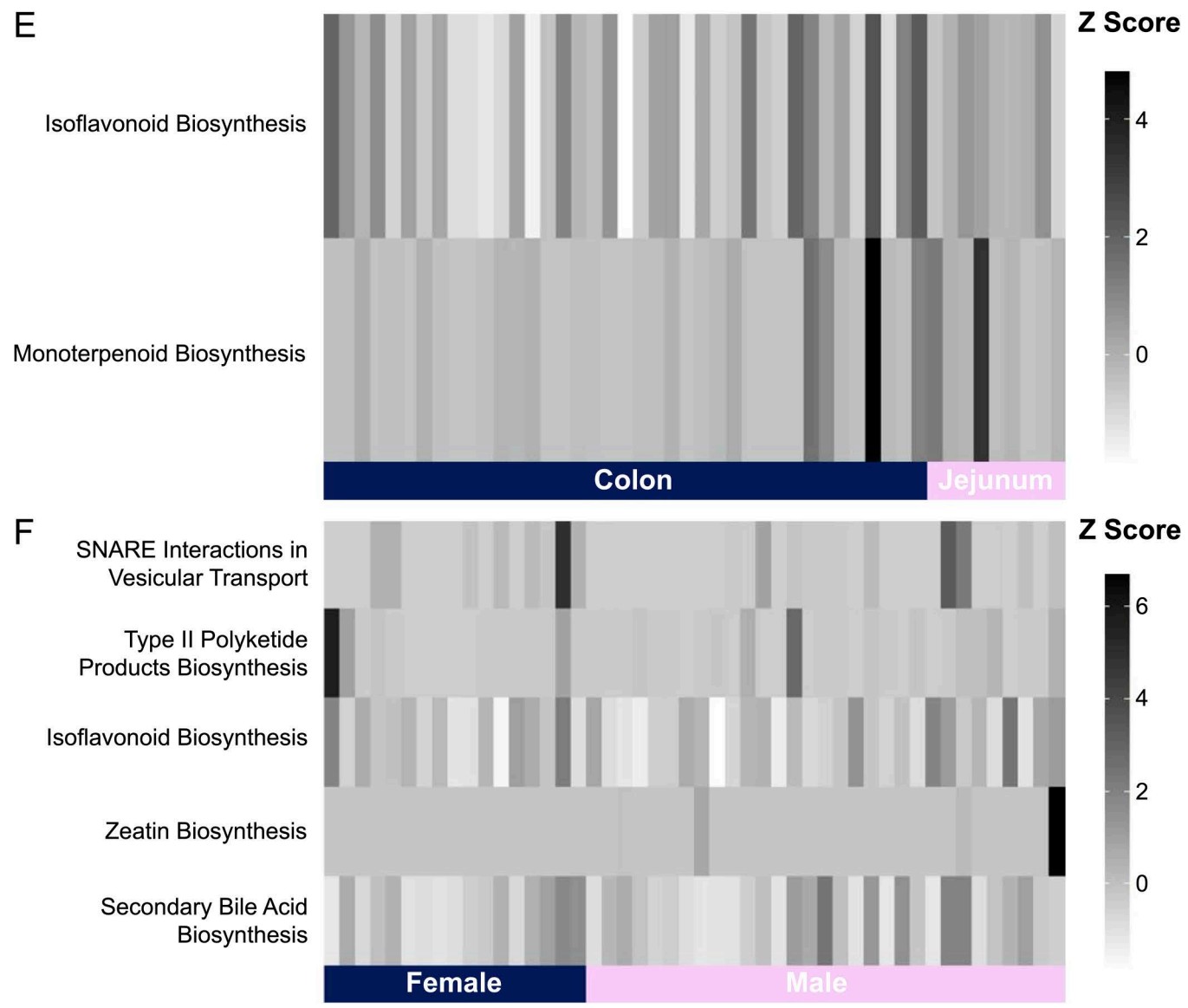

**Fig 10. Heatmaps comparing enrichment patterns of the most relevant pathways between gut sites (A) and sexes (B).** Metabolic pathways were predicted from 16S amplicon sequences for 48 wild American black bears (*Ursus americanus*) harvested in eastern North Carolina, using PICRUSt2.

macronutrient metabolism pathways – and, conversely, how the bear gut microbiome facilitates digestion of a wide range of foods. Shotgun metagenomics also has the potential to identify antimicrobial resistance genes, which are increasingly relevant and impactful to wildlife sharing habitat with human populations.

## Conclusions

GMBs are important to the maintenance of host health. Here we detected nuanced effects of age, gut site and sex on gut microbial membership and putative functionality, despite high levels of inter-individual variation. Understanding carniv-oran GMBs, and black bear GMBs in particular, is necessary to facilitate effective wildlife management and to assess the

impacts of anthropogenic change. Black bears have seasonal foraging strategies, meaning that their GMBs are likely to reflect environmental variation. Future research could investigate gut microbial membership and functionality as it relates to black bears with minimal or no exposure to anthropogenic foods to determine if differences exist with black bears that are exposed to these foods. This would provide information on whether anthropogenic foods are influencing changes in GMB and potential impact on black bear health. In addition, future research should examine GMBs based on patterns of natural seasonal and anthropogenic shifts in resource availability.

## Materials and methods

### Study area

The study area includes Hyde (3,778.9 km$^2$; 58% water) and Tyrrell (1546.7 km$^2$; 35% water) counties in eastern North Carolina, which is part of the Coastal Plain region (Fig 11) [66]. The Coastal Plain consists of relatively flat topography with numerous marshes, swamps, rivers, and the largest naturally occurring lake in North Carolina (i.e., Lake Mattamuskeet). This region supports diverse agriculture and fishing industry, as well as extensive outdoor recreation, including ecotourism. Moreover, substantial portion of these counties is protected as National Wildlife Refuges (e.g., Pocosin Lakes National Wildlife Refuge, Swanquarter National Wildlife Refuge) that support a variety of wildlife including red wolf (*Canis rufus*), bobcat (*Lynx rufus*), American black bear, American alligator (*Alligator mississippiensis*), bald eagle (*Haliaeetus leucocephalus*) and a diverse assemblage of migratory waterfowl and other birds.

Samples were collected from wild American black bears harvested during the regulated hunting season (November 2022) in Hyde and Tyrrell counties in eastern North Carolina. By limiting our sampling effort to a single season and year, we avoided any confounding effects associated with dietary shifts across seasons. Age was estimated by counting cementum layers from a canine tooth [43]. Using these data, we categorized bears into age-classes (yearling = 1, subadult = 2–3, adult ≥ 4) as previously described [21]. Stable isotope values were derived from guard hairs as previously described [21], and DNA was extracted from jejunum (n = 9) and colon (n = 39) samples using the DNEasy PowerSoil Kit (QIAGEN). We used the manufacturer's protocol with the following modifications: we added an extra heated incubation step to break down proteins and a second elution step to maximize DNA yields, as previously described [67]. DNA yields were quantified using a NanoDrop 2000c (ThermoFischer Scientific, MA, USA) and samples were frozen at -80C prior to shipping. Standardized DNA aliquots were shipped on dry ice to Argonne National Laboratory (Lemont, IL, USA) for 16S v4 amplicon library preparation and paired-end sequencing on the Illumina MiSeq platform. Argonne National Laboratory includes negative polymerase chain reaction controls in every amplification plate, and only proceeds with sequencing

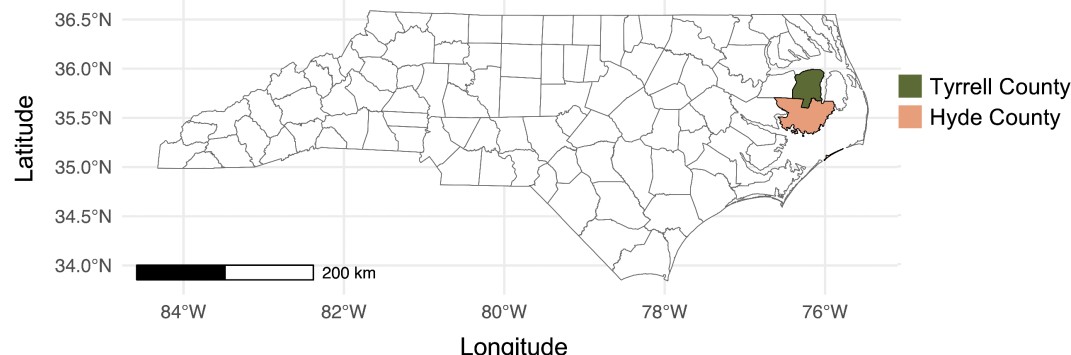

**Fig 11. Study sites of Tyrrell County and Hyde County, North Carolina.** The map was created using the tigris and sf packages in R.

if the negative controls are clean. R code and data are available in the DRYAD repository (https://doi.org/10.5061/dryad.3tx95x6tb).

## Bioinformatic analysis

Raw sequencing data were imported into Quantitative Insights Into Microbial Ecology (QIIME2, version 2019.4) to join sequences, quality-filter, demultiplex, and subsequently call amplicon sequence variants (ASVs) for downstream analysis using the DADA2 QIIME2 plugin [68]. The SILVA 99 database (version 132) was used to classify ASVs to the genus level, then sequences were filtered to remove chloroplasts, mitochondria, Archaea, and any sequences that could not be identified below the kingdom level. We used ranked subsampling normalization [69] to decrease bias in sequencing coverage (Cmin = 24032), retaining all 48 samples.

## Predicting metabolic pathways with PICRUSt2

We used Phylogenetic Investigation of Communities by Reconstruction of Unobserved States (PICRUSt2) to predict the relative abundance of functional pathways from ASV abundance data [65]. Briefly, PICRUSt2 aligns ASVs to reference sequences, then uses ASV abundances to determine gene family abundance and infers pathway abundances from the sample gene family profiles. PICRUSt2 identified 22 of 1009 ASVs that exceeded the maximum NSTI cut-off of 2.0 and were omitted from downstream analyses.

## Statistical analysis

We used R (version 4.2.2) and RStudio (version 2022.12.0 + 353) to analyze ASV tables using the vegan, ggplot2, mctoolsr, rlang, and devtools packages. We used the ggpicrust2 package, an analysis and visualization toolkit designed for working with PICRUSt2 output in R (version 4.41) and RStudio (version 2024.09.0 + 375) [70]. We used the microbial community analysis tools in R (MCtoolsR) package to summarize microbial taxa and create bar charts displaying the dominant phyla (top 6), families (top 10) and genera (top 15). We created heat maps to compare the relative abundance of taxa between gut sites and age classes at the family and genus level. We calculated richness (total number of ASVs) and Shannon Diversity (which takes into account both richness and evenness) to quantify niche space and community complexity within each sample. We used an alpha value of 0.05 as the cutoff for significance.

## Age analyses

We first used linear regression models to quantify the strength and significance of the relationships between weight, richness, and Shannon diversity versus age in years. We next used Kruskal-Wallis tests to compare richness and Shannon diversity across age classes. We used NMDS plots, PERMANOVA (ADONIS2), and SIMPER analyses to evaluate differences in community composition across age classes.

## Gut site comparisons

We compared gut sites across the full dataset, and also created a filtered dataset containing only the paired jejunum and colon samples collected from 9 bears. We compared diversity indices using the Mann Whitney Wilcoxon test, as the sample sizes were small and not normally distributed. We calculated Bray-Curtis distances and created an NMDS plot to visualize clustering patterns and compare community composition for both the full (n = 39 colon and n = 9 jejunum) and paired (n = 9 colon and n = 9 jejunum) data sets. We used PERMANOVA analysis (ADONIS2) to test whether community composition differed significantly by gut site. We also used SIMPER analysis on the full data set to identify which ASVs drove clustering patterns based on Bray-Curtis distances in the NMDS plot. We ran Linear discriminant analysis Effect Size (LEfSe) on paired data to identify significantly and differentially enriched taxa between gut sites.

 

## Male vs female black bears

We compared community composition at the phylum and genus levels in males and females at each gut site. We plotted carbon and nitrogen stable isotope values and compared them to reference values for corn and peanuts. We then compared richness and Shannon diversity between male (n = 5 jejunum, n = 27 colon) versus female (n = 4 jejunum, n = 12 colon) black bears at each gut site using Mann-Whitney-U tests. We used Linear discriminant analysis Effect Size (LEfSe) software [71] to identify significant taxa driving differences between sexes. An LDA score of 2 was used as the threshold for biological relevance [71]. We calculated pairwise Bray-Curtis distances to measure microbial beta diversity and used SIMPER in Rstudio to detect significant bacterial taxa driving patterns in the Bray-Curtis distances. Due to limited sample size, we were unable to run SIMPER for sex within the gut site. We also generated NMDS plots to visualize differences in community composition. Finally, we used Adonis to test for differences in community composition due to sex between the jejunum and colon.

## Predicted metabolic pathways

We used principal component analysis (PCA) of the log-normalized relative abundances of KEGG pathways to assess variation in predicted microbial functional pathways between gut sites and sexes, and created a scree plot to compare the variance accounted for by each principal component (PC). We ran PERMANOVA to test for significant differences in metabolic pathway profiles based on gut site and sex.

We performed differential abundance analysis of predicted KEGG pathways using the "edgeR" method with Benjamini-Hochberg p-value adjustment to compare gut sites (colon vs. jejunum) and sexes (female vs. male). We visualized significantly enriched pathways as bar plots with error bars. We also generated Z-score standardized heatmaps to visualize pathway-level enrichment patterns and variation in the abundance of predicted KEGG pathways across samples by gut site and sex.

## Supporting information

**S1 Table. Bacterial genera driving Bray-Curtis distances among age classes, identified using SIMPER analysis (α = 0.05).** The age of each individual was estimated by counting cementum layers from a canine tooth as described by Marks and Erickson [43]. Using these data, we categorized bears into age-classes (yearling = 1, subadult = 2–3, adult ≥ 4) as previously described [21]. Samples were collected from wild American black bears (*Ursus americanus*) harvested in eastern North Carolina.
(CSV)

**S1 Fig. Scatterplot of weight (kg) versus age (years) for male (n = 19) and female (n = 20) wild American black bears (*Ursus americanus*) harvested in eastern North Carolina.** Age was estimated by counting cementum layers from a canine tooth as described by Marks and Erickson [43].
(TIFF)

**S2 Fig. Scatterplot of Shannon diversity versus age (years) for male (n = 19) and female (n = 20) wild American black bears (*Ursus americanus*) harvested in eastern North Carolina.** Age was estimated by counting cementum layers from a canine tooth as described by Marks and Erickson [43].
(TIFF)

**S3 Fig. Scree plot showing the variance explained by each Principal Component for abundance of predicted metabolic pathways across all 48 samples collected from wild American black bear (*Ursus americanus*).** Metabolic pathways were predicted from 16S amplicon sequences using Phylogenetic Investigation of Communities by Reconstruction of Unobserved States (PICRUSt2).
(TIFF)

## Acknowledgments

We are grateful to Sierra Gillman for processing raw sequencing data.

## Author contributions

**Conceptualization:** Colleen Olfenbuttel, Diana J. R. Lafferty.

**Data curation:** Erin A McKenney, Rachael Hildreth, Colleen Olfenbuttel, Diana J. R. Lafferty.

**Formal analysis:** Erin A McKenney, Erik De Jesus, Taylor Hatfield, Dorian Hayes, Kaleb Holder, Christian Ivarsson, Natalie Morais, Hunter Payne, Kalle Simpson, Adrianna M Staal, Holly Thompson.

**Funding acquisition:** Colleen Olfenbuttel, Diana J. R. Lafferty.

**Investigation:** Rachael Hildreth, Colleen Olfenbuttel, Diana J. R. Lafferty.

**Methodology:** Erin A McKenney, Rachael Hildreth, Colleen Olfenbuttel, Diana J. R. Lafferty.

**Project administration:** Erin A McKenney, Colleen Olfenbuttel, Diana J. R. Lafferty.

**Resources:** Erin A McKenney, Colleen Olfenbuttel, Diana J. R. Lafferty.

**Software:** Erin A McKenney, Erik De Jesus, Taylor Hatfield, Dorian Hayes, Kaleb Holder, Christian Ivarsson, Natalie Morais, Hunter Payne, Kalle Simpson, Adrianna M Staal, Holly Thompson.

**Supervision:** Erin A McKenney, Diana J. R. Lafferty.

**Validation:** Erin A McKenney.

**Visualization:** Erin A McKenney, Erik De Jesus, Taylor Hatfield, Dorian Hayes, Kaleb Holder, Christian Ivarsson, Natalie Morais, Hunter Payne, Kalle Simpson, Adrianna M Staal, Holly Thompson.

**Writing – original draft:** Erin A McKenney, Erik De Jesus, Taylor Hatfield, Dorian Hayes, Kaleb Holder, Christian Ivarsson, Natalie Morais, Hunter Payne, Kalle Simpson, Adrianna M Staal, Holly Thompson.

**Writing – review & editing:** Erin A McKenney, Erik De Jesus, Taylor Hatfield, Dorian Hayes, Kaleb Holder, Christian Ivarsson, Natalie Morais, Hunter Payne, Kalle Simpson, Adrianna M Staal, Holly Thompson, Rachael Hildreth, Colleen Olfenbuttel, Diana J. R. Lafferty.

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
