## [Decision Letter · Decision Letter 0]

11 Nov 2025

Dear Dr. McKenney,

Thank you for submitting your manuscript to PLOS ONE. After careful consideration, we feel that it has merit but does not fully meet PLOS ONE’s publication criteria as it currently stands. Therefore, we invite you to submit a revised version of the manuscript that addresses the points raised during the review process.

We look forward to receiving your revised manuscript.

Kind regards,

Nathan Wolf

Academic Editor

PLOS ONE

Journal Requirements:

https://journals.plos.org/plosone/s/file?id=ba62/PLOSOne_formatting_sample_title_authors_affiliations.pdf....

“Funding was provided to CO by the North Carolina Wildlife Resources Commission and grant funding (W57) through the Federal Aid in Wildlife Restoration Act, popularly known as the Pittman-Robertson Act. In-kind support was provided to DJRL and RH by the College of Arts and Science at Northern Michigan University.”

“Funding was provided to CO by the North Carolina Wildlife Resources Commission and grant funding (W57) through the Federal Aid in Wildlife Restoration Act, popularly known as the Pittman-Robertson Act. In-kind support was provided to DJRL and RH by the College of Arts and Science at Northern Michigan University.”

4. Please be informed that funding information should not appear in the Acknowledgments section or other areas of your manuscript. We will only publish funding information present in the Funding Statement section of the online submission form. Please remove any funding-related text from the manuscript.

5. We note that Figure 9 in your submission contain [map/satellite] images which may be copyrighted. All PLOS content is published under the Creative Commons Attribution License (CC BY 4.0), which means that the manuscript, images, and Supporting Information files will be freely available online, and any third party is permitted to access, download, copy, distribute, and use these materials in any way, even commercially, with proper attribution. For these reasons, we cannot publish previously copyrighted maps or satellite images created using proprietary data, such as Google software (Google Maps, Street View, and Earth). For more information, see our copyright guidelines: http://journals.plos.org/plosone/s/licenses-and-copyright.

a. You may seek permission from the original copyright holder of Figure 9 to publish the content specifically under the CC BY 4.0 license.

We recommend that you contact the original copyright holder with the Content Permission Form (httpp://journals.plos.org/plosone/s/file?id=7c09/content-permission-form.pdf) and the following text:

Additional Editor Comments (if provided):

Thank you for submitting this manuscript to PLOS One. As you will note from the enclosed comments, the Reviewers found the manuscript to be of value and interesting. However, the Reviewers each raised several points that should be considered and potentially revised before the manuscript can be considered for publication.

Reviewers' comments:

Reviewer's Responses to Questions

**Comments to the Author**

1. Is the manuscript technically sound, and do the data support the conclusions?

Reviewer #1: Yes

Reviewer #2: Partly

2. Has the statistical analysis been performed appropriately and rigorously?

Reviewer #1: Yes

Reviewer #2: Yes

3. Have the authors made all data underlying the findings in their manuscript fully available?

Reviewer #1: Yes

Reviewer #2: Yes

4. Is the manuscript presented in an intelligible fashion and written in standard English?

Reviewer #1: Yes

Reviewer #2: Yes

Reviewer #1: In this study, McKenney and colleagues analyzed the gut microbiome composition of 48 black bears that were sampled opportunistically at different sites along the gastrointestinal tract. Using a combination of stable isotope analysis and amplicon sequencing, the authors found that bears did not selectively forage on specific types of bait, and microbial diversity did not vary significantly across gut sites. However, they did observe differences in microbial composition and functional potential across gut regions and between male and female bears. Overall, this study provides valuable samples on wild American black bears and the results are relevant for the field.

Comments:

The figures require refinement to improve clarity and readability. In particular:

• All figures: Add axis labels where missing, clearly indicating what each axis represents and including appropriate units of measurement (e.g., "%"). Please also ensure all figures are in high resolution.

• Figure 1: Please increase the size of the corn/peanut icons to make them more easily visible.

• Figure 2: The legend includes colors that are difficult to distinguish. Please use a more distinguishable color palette where each category is clearly separable from adjacent entries. In alternative, reduce the number of categories shown in the legend.

• Figures 2 and 3: These figures could be merged into a single figure.

• Figures 4, 5, and 6: These plots could also be combined.

• Figures 7 and 8: Consider combining these into a single figure as they are conceptually related. Also, for Figure 7, please increase the axis font size, as the current labels are too small to be easily read.

• Figures 6, 7, and 8: Please use the same colors for males/females used in previous figures (the shades of pink and green/blue in these plots are different).

• Figure 9: Add a brief title to each map panel and include a distance scale bar (e.g., how many miles correspond to 1 cm) to help orient the viewer geographically.

Please provide additional context to the reader on the rationale behind the stable isotopes analysis, and the insights that this kind of analysis provides.

The manuscript would benefit from a deeper discussion of the biological mechanisms that could underlie the observed sex-based differences in microbiome composition and function. Since no sex-specific foraging patterns were detected, the authors should consider and discuss other plausible explanations, such as hormonal differences between males and females that could impact their microbiome.

The authors should also clarify if and to what extent the month in which the samples were collected could explain the observed variability, as different food types are available throughout the year. How was this possible confounding factor considered in the analyses?

The authors may also consider adding that the use of metagenomics has the potential to identify antimicrobial resistance genes, a topic of increasingly relevant interest in the context of wildlife sharing habitat with human populations.

Finally, while the authors have shared both the R code and amplicon data via DRYAD, it is recommended that they also deposit the sequencing data on the European Nucleotide Archive (ENA), as standard practice with microbiome data.

Reviewer #2: McKenney et al 2025

I enjoyed the topic and found the research interesting. It was nicely written, easy to understand and well referenced.

The pdf I was given to review did not have line numbers and there were some formatting issues (irregular line spacing, extra spaces after some words, etc.) that I am not sure were due to the journal’s formatting or the article itself.

This is particularly true with the table and figure titles and makes it harder to review and give feedback

However – there are points below that I believe need to be addressed for clarity and increased value to the community.

Abstract –

General comment – the abstract was clear yet vague. The last sentence, for example, how your findings related to conservation and management is not introduced or explained nor easily deduced from the earlier part of the abstract. (If you want this to be about conservation I think the intro and discussion need to discuss this more and make stronger concrete connections). I think your data is solid and trying to link it to the large conservation piece makes it feel ore flimsy – thus – stick to what you have more IMO.

Introduction

- The NC black bear status is from 2010. Is there not more recent information?

- Some unnecessary repeating in the introduction (re-read for clarity and to eliminate repeating).

- Page 4 - I would personally not use “we” in scientific report although I know this is becoming more common place. Similarly – I would keep the tense the same throughout (moves from past to present)

- “these goals” could be better defined

- Page 5 – above the table could further explain which isotope values

- Page 5 below the table (there is text that I assume is not intended for a find version – says “add new citation from …”

- Pages 5- 6 – the intro regarding bear baiting could / should be simplified and shortened and focus on the current rules that limit baiting in NC & the repeated use of “licensed hunter” seems odd – perhaps define it once and then move on? Also – are these baiting estimates all of NC? I assume differs a bit & even in Terrell Co there are restrictions in which land allows baiting and which doesn’t (thus – where the bear is killed in the Co may also matter)

- Page 6 – when “between” is used to discuss differences between lemur species – should this be “among” thus – more than 2 spots vs just 2?

Materials and Methods

Why is there yellow highlighting on page 20?

Page 22– text varies from saying “the students did …” vs “we did” – please be consistent

Page 23 – your figure 1 didn’t include the meat, etc for these comparisons, right? Why not?

Author Contributions portion – some names are not the same font as others (smaller)

If you want to discuss this population and potential conservation or alterations due to human encroachment, etc – what else can you tell us about these animals? Approximate bear age (data should be available from the teeth) – location found? Weights? Health estimates? Anything? Where they mainly from one of the 2 counties? None of these were from Beaufort Co? Was it private land or public – do we know anything else about these bear?

Results

Sentence 1 – when you state “mid-level” – what is this compared to? Should there be a reference here?

On my version of the pdf figures 1 and 2 and 7 were slightly blurry and the species names particularly hard to read – perhaps a pdf issue only?

Figure 1. The last sentence seems odd on a figure title. I recommend deleting that.

Figure 2 title is in all bold – figure 1 isn’t. Please edit to what the journal wants and be consistent. Also – I like that you note the Eastern NC part in the titles (you don’t do that in all the figures and I would add it to the ones you don’t do it on).

Page 7. Proteo… statement – these differences are statistically different?

Table 2 title ends with “and their respective” which is not clear and needs to be edited

In the pdf for my version – the title for Figure 5 and the text run together and need to be separated - or at least I think that is what happened bc the font is not bold for all of it and the 2nd part doesn’t fit as part of a title. However – the same thing happens for Tables 3 & 4 and thus I am not sure if this is a pdf issue or a title issue. Continuing on this theme of potential inconsistencies – Table 5’s title is a different font size (as are some figures).

Table 3 could be edited so that some of the words are not on a separate line (making it landscape or just making the columns larger for some and smaller for others)

Page 13 – tense is not consistent

Figure 6 has only 8 jejunum samples – I assume the missing sample is discussed somewhere else later? Perhaps discuss here in the results as all for missing samples?

Discussion

Sentence one – I would keep this more generic – we do not know that all these animals had enough access to selectively forage – some may have had peanut access and others may not have (could even be that none of them did).

Page 15 – The sentence “similar to previous carnivore studies – ref 16” doesn’t make sense as this is not a carnivore reference – do you mean carnivore taxonomically or dietarily? Be clear with these discussions and that may help here as wel. In the same paragraph dogs and cats were discussed as being similar but I would consider altering the cat part out as they are carnivores and you are discussing omnivores dietarily. Double check that the references are correct in this section, please. For example – when using the reference 13 (about martens) – should this sentence have other refs as well?

Prior to ref 33 make sure you are saying the potential dispersal of antibiotic resistance (this was said in the prior sentence but in the 2nd usage it makes it sound like they are definitely dispersing antibiotic resistant strains, and this is not known or studied here). Future research statement is vague and I would suggest deleting it.

Page 16 – you not that body weight gain could be more positive for wildlife – but – it should not be assumed always negative in humans – if that is what you mean in the prior sentence then I would alter that sentence and add a reference.

Page 17 – the ref 11 as “our expectations” seems odd as written – it seemed like you meant current expectations not ones from a paper 5 years before –

Discussing age as a limiting factor would be good as well – do you know the ages of any of them? I assume you could get that info from the bear teeth that were collected? I haven’t read the M&M section but also discussing if these were all collected during the same hunting season (as there are two) and in the same year could be potential limitations, etc.

References

Refs are not formatted consistently. Some odd spacing (again – maybe pdf issues – but if not – then fix the punctuation throughout).

Many do not have pages listed. Some have the paper titles having each word with a capital and some just the 1st word. Some abbreviations are not the same throughout.

Species names should be italicized. Does the journal not want doi’s listed?

Does this journal allow personal communication? How do 23 and 27 differ?

Authors for #42

.

Reviewer #1: No

Reviewer #2: No

---

## [Author Response · Author response to Decision Letter 1]

21 Jan 2026

Dear Dr. Wolf,

On behalf of my co-authors, I am pleased to submit our revised research manuscript titled “Gut site and sex-specific enrichment of bacterial taxa and predicted metabolic pathways in wild American black bear (Ursus americanus)” for exclusive publication in PLOS ONE. The gut microbiome both complements and shapes host metabolism and health, and provides valuable insights to inform the management of wildlife populations. Black bears’ omnivorous diets, coupled with simple guts and rapid transit times, result in high inter-individual variation that likely supports their generalist niche and adaptive flexibility, and reflects changes in landscape and foraging. Yet we currently lack understanding of how anthropogenic change may impact black bear gut microbiomes in the southeastern United States. To address that gap, students enrolled in an advanced quantitative course on Gut Microbial Ecology at NC State University partnered with researchers at Northern Michigan University and the North Carolina Wildlife Resources Commission. This cross-institutional collaboration not only increases scientific understanding but provides a unique opportunity for students to apply foundational concepts to wildlife management through an authentic course-based research experience. In this study, we examine differences in stable isotope values, gut microbial community composition and predicted metabolic functions among 48 black bears legally harvested in eastern North Carolina.

We are glad that both reviewers “found the manuscript to be of value and interesting”. “It was nicely written, easy to understand and well referenced.” “Overall, this study provides valuable samples on wild American black bears and the results are relevant for the field.” We have addressed all reviewer concerns below (in bold), including additional analyses to evaluate differences in community composition associated with age and weight. We hope you will now find our manuscript acceptable for publication in PLOS ONE.

We have amended our Funding Statement and Role of Funder Statement as requested:

● Funding was provided by the North Carolina Wildlife Resources Commission and grant funding (W57) through the Federal Aid in Wildlife Restoration Act, popularly known as the Pittman-Robertson Act. In-kind support was provided to DJRL and RH by the College of Arts and Science at Northern Michigan University. There was no additional external funding received for this study. The funders had no role in study design, data collection and analysis, decision to publish, or preparation of the manuscript.

Thank you for considering our work, which includes substantial contributions from undergraduate (E.D.J., D.H., K.H., N.M., H.P., A.M.S., H.T.) and graduate students (T.H., C.I., K.S.), faculty (E.A.M., R.H., D.J.R.L.), and a state regulatory agency (C.O.). Please address all correspondence regarding this manuscript to me at eamckenn@ncsu.edu

Sincerely,

Erin McKenney, MS, PhD (she/they)

Assistant Professor | Director of Undergraduate Programs

Department of Applied Ecology

North Carolina State University

PONE-D-25-36810

Gut site and sex-specific enrichment of bacterial taxa and predicted metabolic pathways in wild American black bear (Ursus americanus)

PLOS ONE

Dear Dr. McKenney,

Thank you for submitting your manuscript to PLOS ONE. After careful consideration, we feel that it has merit but does not fully meet PLOS ONE’s publication criteria as it currently stands. Therefore, we invite you to submit a revised version of the manuscript that addresses the points raised during the review process.

● A rebuttal letter that responds to each point raised by the academic editor and reviewer(s). You should upload this letter as a separate file labeled 'Response to Reviewers'.

● A marked-up copy of your manuscript that highlights changes made to the original version. You should upload this as a separate file labeled 'Revised Manuscript with Track Changes'.

● An unmarked version of your revised paper without tracked changes. You should upload this as a separate file labeled 'Manuscript'.

We look forward to receiving your revised manuscript.

Kind regards,

Nathan Wolf

Academic Editor

PLOS ONE

Journal Requirements:

“Funding was provided to CO by the North Carolina Wildlife Resources Commission and grant funding (W57) through the Federal Aid in Wildlife Restoration Act, popularly known as the Pittman-Robertson Act. In-kind support was provided to DJRL and RH by the College of Arts and Science at Northern Michigan University.”

● Done

● Done

“Funding was provided to CO by the North Carolina Wildlife Resources Commission and grant funding (W57) through the Federal Aid in Wildlife Restoration Act, popularly known as the Pittman-Robertson Act. In-kind support was provided to DJRL and RH by the College of Arts and Science at Northern Michigan University.”

● Done

● Done

4. Please be informed that funding information should not appear in the Acknowledgments section or other areas of your manuscript. We will only publish funding information present in the Funding Statement section of the online submission form. Please remove any funding-related text from the manuscript.

● Done

5. We note that Figure 9 in your submission contain [map/satellite] images which may be copyrighted. All PLOS content is published under the Creative Commons Attribution License (CC BY 4.0), which means that the manuscript, images, and Supporting Information files will be freely available online, and any third party is permitted to access, download, copy, distribute, and use these materials in any way, even commercially, with proper attribution. For these reasons, we cannot publish previously copyrighted maps or satellite images created using proprietary data, such as Google software (Google Maps, Street View, and Earth). For more information, see our copyright guidelines: http://journals.plos.org/plosone/s/licenses-and-copyright.

● We have created a replacement figure in R, and updated the code in our DRYAD submission to include the script used to make the map.

a. You may seek permission from the original copyright holder of Figure 9 to publish the content specifically under the CC BY 4.0 license.

We recommend that you contact the original copyright holder with the Content Permission Form (httpp://journals.plos.org/plosone/s/file?id=7c09/content-permission-form.pdf) and the following text:

Additional Editor Comments (if provided):

Thank you for submitting this manuscript to PLOS One. As you will note from the enclosed comments, the Reviewers found the manuscript to be of value and interesting. However, the Reviewers each raised several points that should be considered and potentially revised before the manuscript can be considered for publication.

Reviewers' comments:

Reviewer's Responses to Questions

Comments to the Author

1. Is the manuscript technically sound, and do the data support the conclusions?

Reviewer #1: Yes

Reviewer #2: Partly

2. Has the statistical analysis been performed appropriately and rigorously?

Reviewer #1: Yes

Reviewer #2: Yes

3. Have the authors made all data underlying the findings in their manuscript fully available?

Reviewer #1: Yes

Reviewer #2: Yes

4. Is the manuscript presented in an intelligible fashion and written in standard English?

Reviewer #1: Yes

Reviewer #2: Yes

5. Review Comments to the Author

Reviewer #1: In this study, McKenney and colleagues analyzed the gut microbiome composition of 48 black bears that were sampled opportunistically at different sites along the gastrointestinal tract. Using a combination of stable isotope analysis and amplicon sequencing, the authors found that bears did not selectively forage on specific types of bait, and microbial diversity did not vary significantly across gut sites. However, they did observe differences in microbial composition and functional potential across gut regions and between male and female bears. Overall, this study provides valuable samples on wild American black bears and the results are relevant for the field.

Comments:

The figures require refinement to improve clarity and readability. In particular:

• All figures: Add axis labels where missing, clearly indicating what each axis represents and including appropriate units of measurement (e.g., "%"). Please also ensure all figures are in high resolution.

• Figure 1: Please increase the size of the corn/peanut icons to make them more easily visible.

Done.

• Figure 2: The legend includes colors that are difficult to distinguish. Please use a more distinguishable color palette where each category is clearly separable from adjacent entries. In alternative, reduce the number of categories shown in the legend.

We have reduced the number of taxa in the legend as suggested.

• Figures 2 and 3: These figures could be merged into a single figure.

We appreciate your suggestion. However, Reviewer 2 requested that we re-analyze the gut microbial data based on bear weights and ages. As such, we have re-worked the original figures (as well as the Methods, Results, and Discussion) to incorporate those new analyses and findings. Figures 2 now contains bar charts for age classes as well as gut sites.

• Figures 4, 5, and 6: These plots could also be comb

---

## [Decision Letter · Decision Letter 1]

5 Mar 2026

Gut site and sex-specific enrichment of bacterial taxa and predicted metabolic pathways in wild American black bear (Ursus americanus)

PONE-D-25-36810R1

Dear Dr. McKenney,

We’re pleased to inform you that your manuscript has been judged scientifically suitable for publication and will be formally accepted for publication once it meets all outstanding technical requirements.

Kind regards,

Nathan Wolf

Academic Editor

PLOS One

Additional Editor Comments (optional):

Thank you for the robust responses to the Reviewer's concerns. The resulting manuscript is very strong and a great contribution to PLOS One.

Reviewers' comments:

Reviewer's Responses to Questions

**Comments to the Author**

Reviewer #1: All comments have been addressed

Reviewer #2: All comments have been addressed

2. Is the manuscript technically sound, and do the data support the conclusions?

Reviewer #1: Yes

Reviewer #2: Yes

3. Has the statistical analysis been performed appropriately and rigorously?

Reviewer #1: Yes

Reviewer #2: Yes

4. Have the authors made all data underlying the findings in their manuscript fully available?

Reviewer #1: Yes

Reviewer #2: Yes

5. Is the manuscript presented in an intelligible fashion and written in standard English?

Reviewer #1: Yes

Reviewer #2: Yes

Reviewer #1: (No Response)

Reviewer #2: The authors addressed every comment that I made (and I made a lot). I have no further edits and I am excited for the PI and the students to have their work published.

.

Reviewer #1: No

Reviewer #2: No

---

## [Editor Report · Acceptance letter]

PONE-D-25-36810R1

PLOS One

Dear Dr. McKenney,

I'm pleased to inform you that your manuscript has been deemed suitable for publication in PLOS One. Congratulations! Your manuscript is now being handed over to our production team.

Kind regards,

on behalf of

Dr. Nathan Wolf

Academic Editor

PLOS One